# Evaluation of Tidal Effect in Long-Strip DInSAR Measurements Based on GPS Network and Tidal Models

**Wei Peng** [1,2], **Qijie Wang** [3,4,*], **Yunmeng Cao** [5], **Xuemin Xing** [1] and **Wenjie Hu** [1]

1. Hunan International Scientific and Technological Innovation Cooperation Base of Advanced Construction and Maintenance Technology of Highway, Changsha University of Science & Technology, Changsha 410114, China; pengwei@csust.edu.cn (W.P.); xuemin.xing@csust.edu.cn (X.X.); huwenjie@stu.csust.edu.cn (W.H.)
2. School of Traffic & Transportation Engineering, Changsha University of Science & Technology, Changsha 410114, China
3. School of Geosciences and Info-Physics, Central South University, Changsha 410083, China
4. Hunan Key Laboratory of Remote Sensing of Ecological Environment in Dongting Lake Area, Changsha 410007, China
5. GNS Science, Lower Hutt 5040, New Zealand; y.cao@gns.cri.nz
* Correspondence: qjwang@csu.edu.cn; Tel.: +86-13808425350

**Abstract:** A long-strip differential interferometric synthetic aperture radar (DInSAR) measurement based on multi-frame image mosaicking is currently the realizable approach to measure large-scale ground deformation. As the spatial range of the mosaicked images increases, the nonlinear variation of ground ocean tidal loading (OTL) displacements is more significant, and using plane fitting to remove the large-scale errors will produce large tidal displacement residuals in a region with a complex coastline. To conveniently evaluate the ground tidal effect on mosaic DInSAR interferograms along the west coast of the U.S., a three-dimensional ground OTL displacements grid is generated by integrating tidal constituents' estimation of the GPS reference station network and global/regional ocean tidal models. Meanwhile, a solid earth tide (SET) model based on IERS conventions is used to estimate the high-precision SET displacements. Experimental results show that the OTL and SET in a long-strip interferogram can reach 77.5 mm, which corresponds to a 19.3% displacement component. Furthermore, the traditional bilinear ramp fitting methods will cause 7.2~20.3 mm residual tidal displacement in the mosaicked interferograms, and the integrated tidal constituents displacements calculation method can accurately eliminate the tendency of tidal displacement in the long-strip interferograms.

**Keywords:** ground tidal deformation; differential InSAR; GPS; tidal models

## 1. Introduction

In coastal interferometric synthetic aperture radar (InSAR) deformation measurements, the ground tidal displacements along the LOS direction can reach 2~4 decimeters, and it usually reaches centimetres in a single-frame differential InSAR (DInSAR) interferogram with a spatial range of 100~250 km [1]. Currently, the tidal effects are generally ignored in InSAR deformation measurements, while a best-fitting ramp is applied to remove the spatial residual large-scale errors in differential interferograms [2–4]. The traditional bilinear ramp fitting method can eliminate tidal displacements in most instances, such as inland areas or small-range DInSAR measurements because the magnitude of the ocean tidal loading (OTL) displacement within the SAR image is minor and the spatial variations of the solid earth tide (SET) displacements tend to be linear ramps. However, with the increasing spatial range of a long-strip differential interferogram based on mosaicked multi-frame images, the SET displacement can reach 3 decimeters, and the OTL displacement from the coastline to inland reaches several centimetres with significant nonlinear variations in complex coastline areas. If the bilinear ramp fitting method is applied to the coastal

long-strip interferograms, the residual of the tidal displacement can be up to centimetre-level [5,6]. Therefore, the long-strip DInSAR measurements covering the coastal areas with significant tidal variation need to initially be assessed [7].

The ocean tidal models and GPS precise point positioning (PPP) methods are the most commonly used analysis and correction methods for studies of ground OTL displacements in InSAR measurements [8]. Without considering the atmospheric delay and other large-scale errors, DiCaprio et al. simulated the OTL displacements in the ERS and ENVISAT data based on the FES2004 ocean tidal model, demonstrating the importance of correcting the nonlinear OTL displacement in a DInSAR interferogram [9,10]. If atmospheric delay and other large-scale errors are considered, they need to be settled by external correction data or two-dimensional image separation methods, which are limited to the accuracy of atmospheric delay and other large-scale error correction [11,12]. Yu et al. simulated the OTL displacement in global Sentinel-1 data based on the FES2014b model and analysed the comprehensive impact of the OTL effect and atmospheric delay error [13]. Wu et al. compared the differences of the ocean tide models in OTL correction of the InSAR measurements [14]. However, the global ocean tide models are ocean grids with a low spatial resolution (0.125° or 0.5°) [10,15], which need Green's function to point by point inefficiently calculate the OTL displacements of the loading point [16,17]. For the tidal effect evaluation of the DInSAR measurements with a large number of pixels, obtaining spatial high-precision and high spatial-resolution coastal ground tidal displacement grid is extremely important.

According to tidal theory, the SET effect is relevant to the tidal force and the Earth's internal structure, and it can be accurately modelled using the SET model of the 2010 IERS Convention [18,19]. In contrast, the expressions between OTL displacement and ocean tidal height, geographical location and coastline shape are more complicated, and the spatial OTL displacement based on this method needs to be further validated [20]. The OTL displacement can generally be decomposed into eight main tide constituents (semi-diurnal constituents M2, N2, S2, K2 and diurnal constituents Q1, O1, P1, K1) according to the harmonic analysis method, and the GPS precise point positioning (PPP) technique has been shown to accurately determine the tide constituents displacement of semi-diurnal principal lunar M2, semi-diurnal elliptical lunar N2, diurnal principal lunar O1 and diurnal elliptical lunar Q1 in the north (N), east (E) and up (U) directions [21–23]. As a dense globally distributed OTL displacement measurement method, it has been used to evaluate the spatial accuracy of the OTL models [24]; on the other hand, it can be used to improve the spatial OTL displacements in complex coastline areas. Therefore, the OTL displacement in the three-dimensional directions can be estimated from the PPP time series and tidal models, taking these high-precision OTL displacements and SET displacements based on the SET model, we proposed a three-dimensional ground tidal displacement grid, which can convert to any line of sight (LOS) direction of pixels in the differential interferogram to analyse the ground tidal deformation.

In this paper, Section 2 introduces the acquisition method of the high-precision three-dimensional tidal displacements, Section 3 introduces the evaluation and correction of the tidal displacement in mosaicked long-strip differential interferograms using the tidal estimation of the kinematic PPP and tidal models.

## 2. Materials and Methods

### 2.1. Traditional Tidal Models

The OTL displacement is usually calculated by the convolution integral of Green's loading function and the global ocean tidal height provided by ocean tidal models. The main principle is to use Green's function to describe the superimposition of the deformation field on the loading point caused by the global ocean tidal height variation, which can be expressed as [25]:

$$d^{OTL\_model}(L, B, t) = \iint_S \rho R^2 H(L', B', t) G(\theta, \beta) \sin\theta \, d\theta \, d\beta \tag{1}$$

where $d^{OTL\_model}(L, B, t)$ is the ocean tide displacement of the model at the calculation point, $\rho$ is the seawater density, $R$ is the radius of the earth, and $L'$ and $B'$ are the spherical coordinates of the loading point, respectively. $H(L', B', t)$ is the instantaneous tide height provided by the tide models. $G(\theta, \beta)$ is Green's function, and $\theta$ and $\beta$ are the spherical distance and azimuth between the calculation point and the loading point, respectively. The spatial estimation accuracy of this traditional method strongly depends on the spatial resolution and accuracy of the tide model, which still has deficiencies. Therefore, it is necessary to determine its spatial estimation accuracy in complex coastline areas.

### 2.2. Spatiotemporal Modelling of OTL Displacement Estimated from PPP Time Series

According to the 2010 IERS Convention, the OTL displacement consisting of *M* tidal constituents can be written as a tidal harmonic function:

$$d^{OTL}_{m,n}(t) = \sum_{m=1}^{M} f_m A_m \cos(\omega_{m,n}t + \chi_m + \mu_m - \Phi_{m,n}) \tag{2}$$

where $A_{m,n}$ refers to the amplitude, and $\Phi_{m,n}$ represent phase delay, $\omega_{m,n}$ is the angular frequency, *n* is GPS sites, $f_m$ is the node factor, $\chi_m$ and $\mu_m$ are the initial astronomical angle and astronomical angle, respectively.

Based on Formula (2), the amplitude and phase delay can be estimated from the kinematic PPP coordinate time series of the GPS network, which are spatial location-related parameters that can be constructed as a phasor:

$$Phasor_m(x, y) = \begin{bmatrix} Phasor1_m \\ Phasor2_m \end{bmatrix} = \begin{bmatrix} A_m(x, y) \cos(\Phi_m(x, y)) \\ A_m(x, y) \sin(\Phi_m(x, y)) \end{bmatrix} \tag{3}$$

where, $(x, y)$ is the longitude and latitude. In terms of spatial dimension, the observations of GPS reference sites are independent. $Phasor^{PPP}_{m,n}$ modelling follows the spatial tendency of constituents *m* can eliminate spatial random errors and predict high-precision and high-resolution OTL displacements.

$$Phasor^{PPP}_{m,n} = b_0 + f_{LSSVM}\left(Phasor^{Model}_{m,n}\right) \tag{4}$$

where $f_{LSSVM}$ is the least squares support vector machine (LSSVM) based on the polynomial kernel function [26], $b_0$ is a constant deviation, and $Phasor^{Model}_{m,n}$ is the phasor of the *m* tidal constituents of the OTL model at the location of *n* GPS sites with the order from large to small, and the tidal constituents displacements of any tide loading point *p* in the range of the GPS reference sites network can be predicted. In coastal areas with a high site density of GPS networks, the expression of the site location and the constituents can be determined based on the Formula (4).

After that, the spatiotemporal modelled M2, N2, O1 and Q1 tidal constituent displacements of a GPS reference station network and P1, K1 and K2 tidal constituent displacements of an ocean tidal model are composed to obtain spatial higher precision OTL displacements. According to Equations (2) and (3), the tidal constituents' displacement is determined based on a GPS reference sites network and a tidal model can be blended to estimate the OTL displacement $d^{OTL}$ measured in long-strip differential interferograms.

$$d^{OTL}_{m_1+m_2,p}(t) = \sum_{m_1=1}^{4} \begin{bmatrix} B^1_{m_1} & B^2_{m_1} \end{bmatrix} \begin{bmatrix} Phasor1^{PPP}_{m_1,p} \\ Phasor2^{PPP}_{m_1,p} \end{bmatrix} + \sum_{m_2=1}^{3} \begin{bmatrix} B^1_{m_2} & B^2_{m_2} \end{bmatrix} \begin{bmatrix} Phasor1^{Model}_{m_2,p} \\ Phasor2^{Model}_{m_2,p} \end{bmatrix} \tag{5}$$

$$\begin{aligned} B^1 &= f \cos(\omega t + \chi + \mu) \\ B^2 &= f \sin(\omega t + \chi + \mu) \end{aligned} \tag{6}$$

where, $m_1$ is Q1, O1, N2, and M2 constituents estimation of a GPS reference station network, $m_2$ is P1, K1, and K2 constituents estimation of a global/regional ocean tidal model, $B^1$ and $B^2$ are constants in the spatial domain.

### 2.3. Tidal Displacements in the Long-Strip Differential Interferogram

Traditional InSAR measurements believe that the spatial large-scale ground tide effect can be ignored in the course of multiple imaging ground deformations measured by SAR satellite systems and that the spatial relative distance between the sensor and the target object in an InSAR image does not affect by the tidal deformation of the OTL and SET. Based on this, the usual differential interference model does not consider the ground tidal phases, and its displacement expression of the pixel in the LOS direction of the differential interferogram can be expressed as [27]:

$$d_{x,y} = d_{x,y}^{defo} + d_{x,y}^{atmospheric} + d_{x,y}^{topo} + d_{x,y}^{orbit} + \varepsilon \tag{7}$$

where $x$ and $y$ are the longitude and latitude of pixels in the interferogram after geocoding, $d_{x,y}^{defo}$, $d_{x,y}^{atmospheric}$, $d_{x,y}^{topo}$, $d_{x,y}^{orbit}$, and $\varepsilon$ are the ground deformation, atmospheric delay error, topography residual error, orbital error, and other residual signals, respectively. To analyse the ground tidal displacement, the interferogram without spatial large-scale land subsidence and earthquakes were selected based on the PPP coordinate time series of the GPS sites network, so the $d_{x,y}^{defo}$ mainly contains the ground tidal deformation in the coastal areas. The atmospheric delay error $d^{atmospheric}$ can be reduced by using the ICAMS advanced atmospheric correction method based on the ECMWF ERA-5 global atmospheric model [28]. If the ground tidal deformation is not considered, the residual large-scale error is usually believed to approach the linear plane, and fitting these errors with a bilinear function model yields:

$$d_{x,y} - d_{x,y}^{atmospheric} = d_{x,y}^{large-scale\_errors} + \varepsilon = a_0 + a_1 x + a_2 y \tag{8}$$

where $a_0$, $a_1$, $a_2$ are the coefficients to be sought to fit the plane. For the multi-frame mosaic interferograms in the coastal ground areas, it is erroneous to eliminate the spatial nonlinear ground tidal deformation by the bilinear function. Considering the ground tidal displacement, the large-scale error fitting model can be expressed as:

$$d_{x,y} - d_{x,y}^{atmospheric} - d_{x,y}^{OTL} - d_{x,y}^{SET} = d_{x,y}^{large-scale\_errors} + \varepsilon = a_0 + a_1 x + a_2 y \tag{9}$$

where the SET displacement $d_{x,y}^{SET}$ can be calculated by using the SET method in the 2010 IERS Convention, its accuracy is submillimetre, and the OTL displacement $d_{x,y}^{OTL}$ can be calculated and mutually validated by the GPS network and ocean tidal models.

### 2.4. Tidal Data Analysis and Processing

The global OTL displacements are usually calculated using an ocean tidal model-based loading Green's function method. Taking the FES2014b model as an example, the up (U), north (N) and east (E) OTL displacements $d_{n,p}^{OTL}$ of point $p$ with an interval of 300 s in the year 2019 can be obtained by using the ocean tidal model-based loading Green function method, and the variation of the OTL displacements in three-dimension is evaluated by the standard deviation (StdDev) $S_p^{OTL}$, which can be expressed as [12]:

$$S_p^{OTL} = \sqrt{\frac{1}{N-1} \sum_{n=1}^{N} \left( d_{n,p}^{OTL} - \overline{d_p^{OTL}} \right)^2} \tag{10}$$

where $N$ is sample numbers of OTL displacement time series, $\overline{d_p^{OTL}}$ is the average of OTL displacement time series. According to the OTL StdDev values of $p$ pixels in spatial large-

scale differential interferograms, the magnitude and spatial relative variation characteristics of the ground OTL effect can be preliminarily evaluated. The OTL displacement varies greatly in some coastal areas around the world, including the west coast of the United States, the west coast of Europe, the northeast coast of South America, and the south coast of Africa. (in the red box in Figure 1).

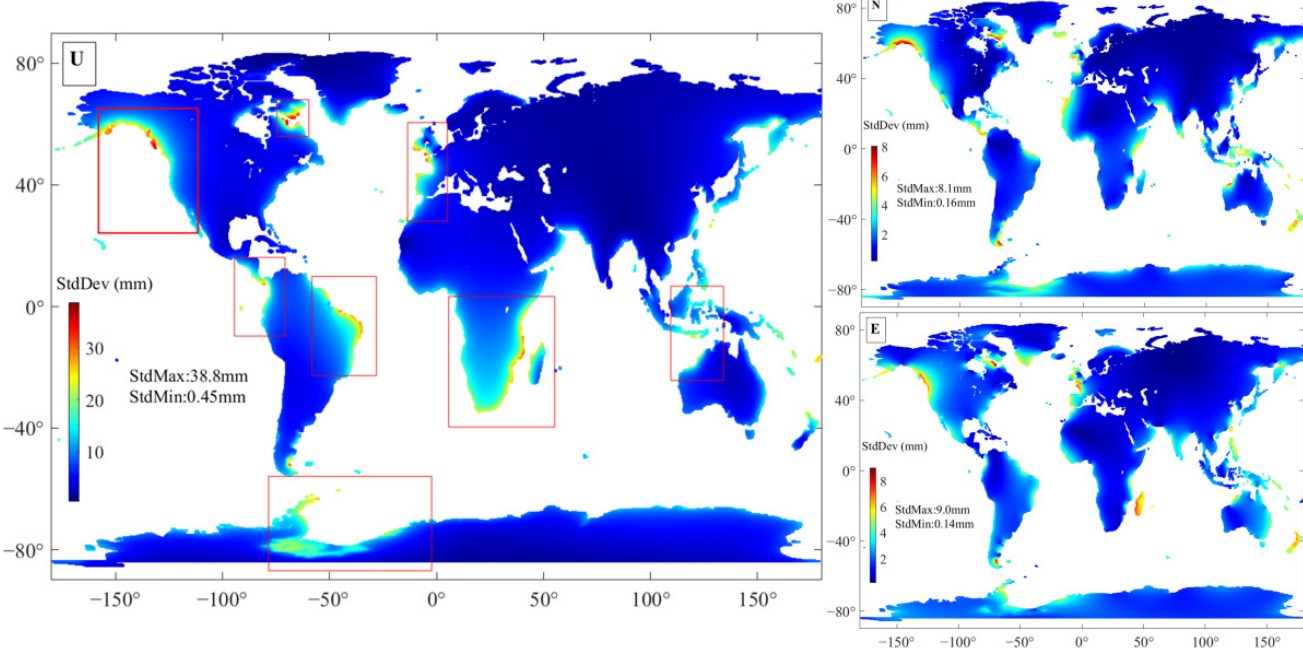

**Figure 1.** The standard deviation map of global ground OTL displacement with an interval of 300 s in the U, N, and E directions based on the FES2014b model.

Due to a large amount of Sentinel-1 SLC ascending data, the dense number of GPS continuous reference stations and a larger OTL effect on the west coast of the U.S., the tidal effect in long-strip DInSAR measurements can be evaluated using the tidal estimations of the GPS network and tidal models. To measure the larger-range ground deformation, nine-frame Sentinel-1 SLC images with a range of 250 km on track 137 were mosaicked as a long-strip differential interferogram (see Figure 2), and the imaging time of the long-strip SAR images was approximately 01:59:30 UTC. The length of the mosaicked image is more than 1600 km, covering a variety of terrains such as the ocean and flat and mountainous areas. There is lower coverage of forest in the southern area, and the coherence of SAR images is high, which is often used to analyse earthquakes, landslides, and other geological tectonic activities. The specific InSAR data processing is as follows:

(i).   Image registration, interferogram generation, removal of the flat phase using SRTM with a 30 m resolution, phase unwrapping follows the minimum cost flow method, and geocoding is performed on the Sentinel-1 SLC image using GAMMA software [29]. The precise orbital file is added in the data processing, and the plane fitting is not implemented until the ground tidal displacement correction.

(ii).  The 29 differential interferograms are selected based on the principle of a small baseline [30].

(iii). The differential interferograms of the nine adjacent frames are mosaicked to obtain the long-strip differential interferograms.

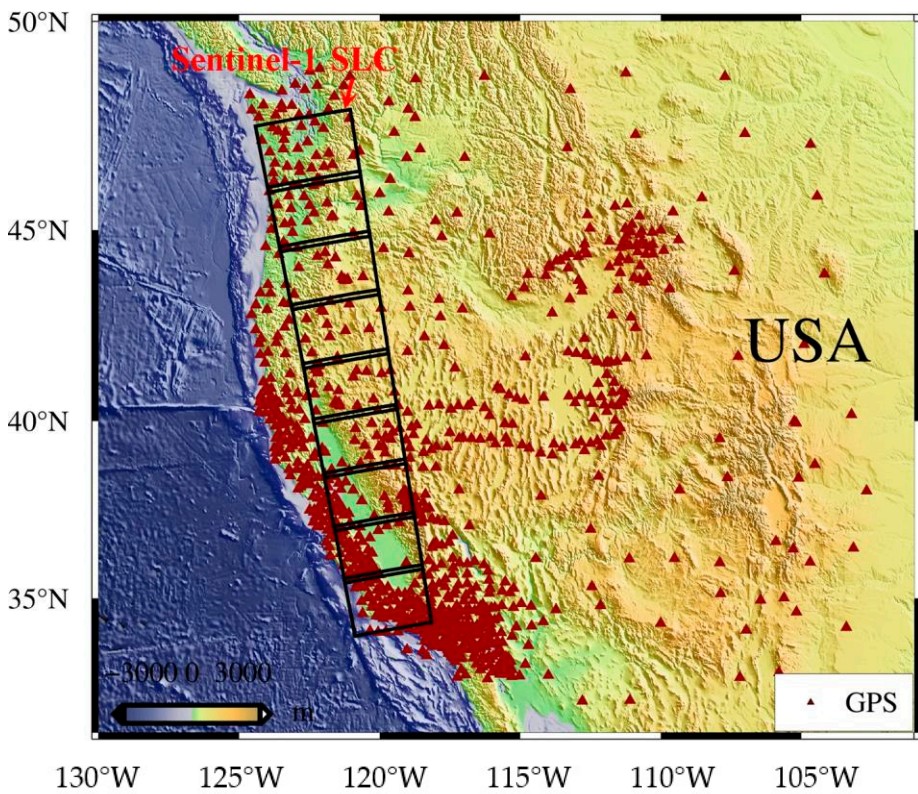

**Figure 2.** Distribution of 1038 continuous GPS sites and multiple Sentinel-1 SLC image ranges on the west coast of the U.S.

In this study, 1038 GPS reference stations of the Plate Boundary Observation (PBO) network were selected (see Figure 2), and the time range of the GPS observations is from the year 2014 to 2019. All observations of 1826 days are processed using the kinematic PPP processing module of Bernese V5.2 to obtain the PPP coordinate time series in the U, N, and E directions with 600 s as the sample interval. The three-dimensional coordinate time series are pre-processed by outlier (>3σ) elimination and wavelet filter denoising, then the ground OTL displacement can be estimated from the kinematic PPP coordinate time series of a GPS reference station network.

## 3. Results

### 3.1. The OTL Estimation Based on Kinematic PPP and Ocean Tide Models

After kinematic PPP data processing, the three-dimensional ground displacement time series is fitted by the tidal harmonic function to obtain the amplitude and phase delay parameters of the tidal constituents, which can be constructed as a phasor. Meanwhile, the offshore grid of the FES2014b global model is replaced by the higher spatial resolution of the regional tidal model osu.usawest, which is compared with the phasor of the tidal constituents measured by kinematic PPP (see Figure 3). The spatial precision of the constituent's phasor estimated based on the kinematic PPP technique is different, where Q1, O1, N2, and M2 tide constituents have higher estimation accuracy, and their spatial trend is consistent with the FES2014b+osu.usawest model. Although the P1, K1 and K2 constituent tendencies are similar to those of the tidal model, the random error in the spatial dimension is large. The SAR satellite's revisit period is an integral multiple of the constituent S2 (12 h) period, the displacement of constituent S2 at multiple imaging times is the same, which is offset in the differential interference.

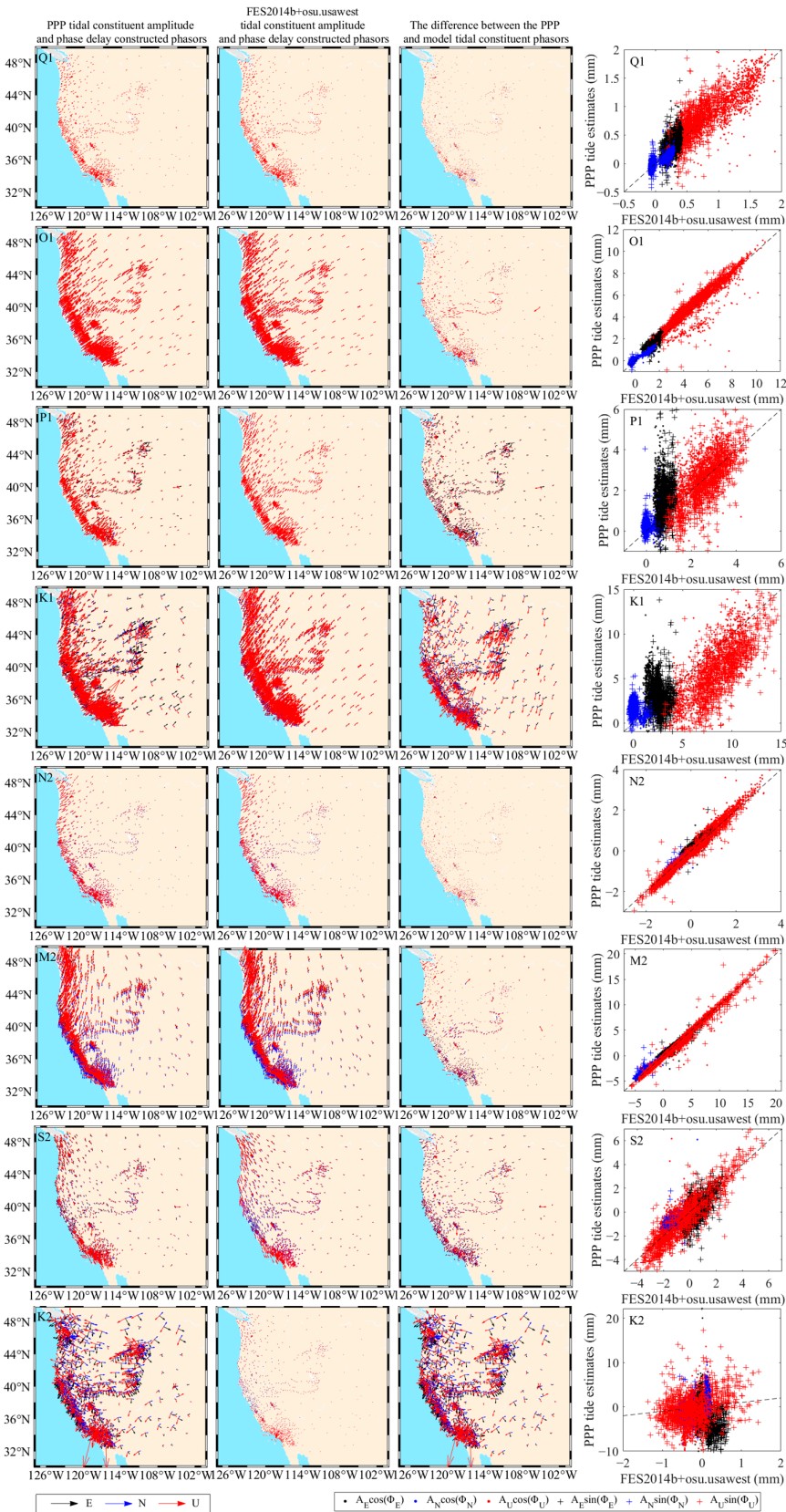

**Figure 3.** Comparative analysis of the phasor of tidal constituents estimated by the kinematic PPP technique and FES2014b+osu.usawest models.

According to the spatial characteristics of each tidal constituent's phasor of the tidal model, the phasor determined by kinematic PPP is fitted by Formula (4), and the predicted Q1, O1, N2, M2 tidal parameters are used to replace the corresponding tidal harmonic parameters of the tide model to form the three-dimensional ground OTL displacements. The StdDev map difference between the proposed tide calculation method and the FES2014b+osu.usawest model is calculated in Figure 4. The coastland areas A and B within the two rectangles in Figure 4 have larger differences between the ground OTL estimations of the proposed OTL calculation method and the FES2014b+osu.usawest model. Its maximum StdDev value of the displacement difference is 1.93 mm, and its corresponding that of the vertical OTL displacement is 17.9 mm, which suggests that the OTL displacement difference between the two methods reach 10.7%. Likewise, the StdDev values of the north direction are 0.37 mm, 3.61 mm, and 10.3%, respectively, and those of the east direction are 0.91 mm, 5.85 mm, and 15.6%. The differential interferograms in Section 3.2 will be used to verify the improvement of the correction accuracy of the proposed method in coastland area A.

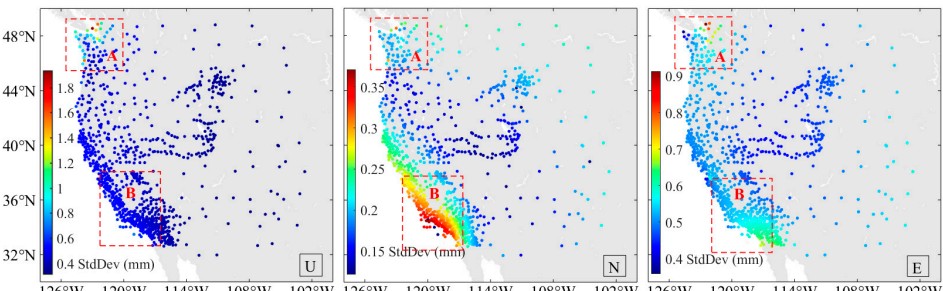

**Figure 4.** StdDev maps of the difference of the OTL displacements estimated by the proposed tide calculation method and FES2014b+osu.usawest model in U, N, and E directions, and the areas with lager difference (red box).

Furthermore, a three-dimensional ground OTL displacements grid with a spatial resolution of $1' \times 1'$ is formed by integrating the spatiotemporal modelled Q1, O1, N2, and M2 constituents of a GPS reference station network and the P1, K1, and K2 constituents of the FES2014b+osu.usawest model. For the long-strip differential interferogram of Sentinel-1 data covering the west coast of the United States, the StdDev values are 1.89 mm, 18.16 mm, and 10.4% in the LOS direction (ascending; the incident angle is 39°; the time interval is 12 days), which means millimetre to centimetre OTL displacement residuals in a differential interferogram and time-series InSAR measurement (see Figure 5).

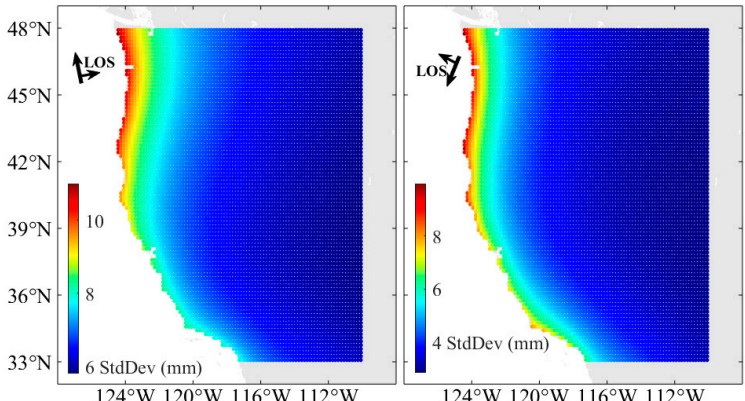

**Figure 5.** The OTL displacement StdDev along the LOS direction (Incident angle is 39°) of the ascending (Heading direction is −13°) and descending (Heading direction is 193°) Sentinel-1 measurements with an interval of 12 days based on the proposed OTL calculation method.

### 3.2. Assessment and Removal of Tide Displacements in a Long-Strip Differential Interferogram

For the 29 long-strip differential interferograms, the spatial variations in OTL and SET and their superposition ground displacement in the long-strip differential interferograms are calculated. The results in Figure 6 reveal that the ground tidal displacements in the long-strip differential interferograms can reach 77.5 mm; the largest SET displacement is 78.9 mm and the largest OTL displacement is 41.9 mm, which illustrates that the magnitude of the tidal displacements can match the atmospheric delay error in some differential interferograms. If the interferograms are fitted by the bilinear ramp, the largest tidal displacement residual generated in interferogram 20180906-20181012 is 20.3 mm and that of interferogram 201800801-20180813 is the smallest, which is 4.9 mm.

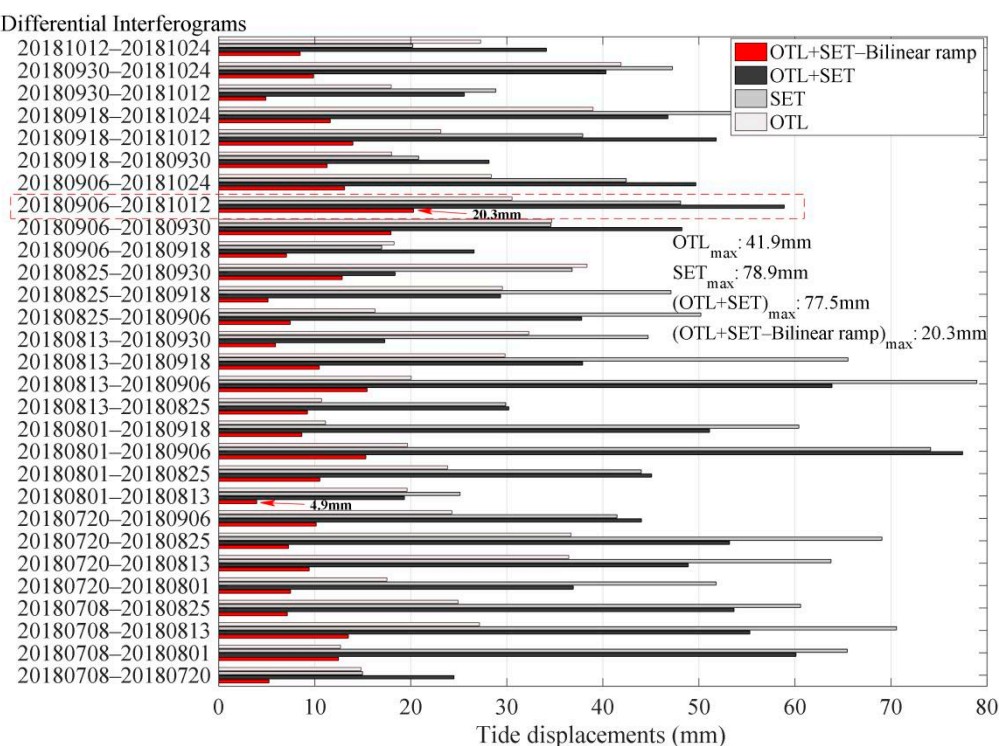

**Figure 6.** The SET and OTL displacements in 29 long-strip differential interferograms and the error analysis of residual tidal displacement produced by bilinear fitting ramp, and the interferogram with largest residual tidal displacement (red box).

According to the comparison of the tidal displacement residuals in Figure 7, the long-strip differential interferogram is fitted by the bilinear fitting method, and the tidal displacement residual in the near coastline area can reach 20.3 mm (area B in Figure 7(a4)) and 7.2 mm using splice bilinear ramp fitting (area A in Figure 7(a5)); therefore, the traditional plane-fitting methods cannot eliminate the SET and OTL displacements. For the area far from the coastline, both plane fitting methods make minor errors (area C in Figure 7(a5)). However, since the tidal displacement residual remains a long-wavelength signal, it has a limited impact on the spatial small-scale differential interferogram (area B in Figure 7(a5)).

In a large-scale spatial signal, the external correction of atmospheric delay error mainly corrects the terrain-related atmospheric delay error [31]. The spatial variation of SET displacement is not affected by the ground geographical environment factors, and its spatial variations are close to the uniform curve plane, while the OTL displacement is affected by the shape of the coastline, and the spatial variation shows the trend of gradual weakening from the coastline to inland, and the other large-scale errors can be neglectable based on ground deformation analysis of the GPS sites network in this study. The atmospheric delay error (especially for long-wavelength and topography-related atmospheric delay errors)

and tidal displacements in the long-strip differential interferograms can be corrected using the atmospheric delay, SET and OTL correction models, and then bilinear ramp fitting was performed to eliminate the other large-scale error residuals (Figure 8(a1–d3)).

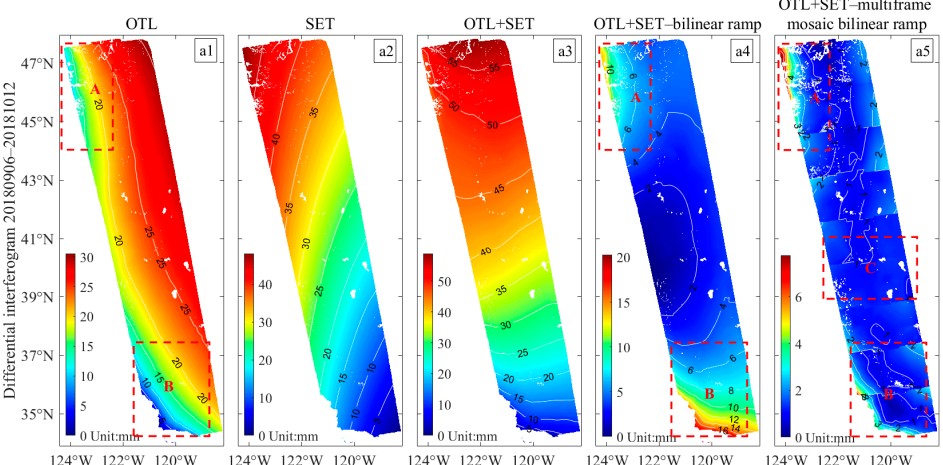

**Figure 7.** OTL displacement (**a1**), SET (**a2**) and their superposition displacement (**a3**) based on the differential interferogram acquisition on 6 September 2018 and 12 October 2018; (**a4**) residual tidal displacement generated by the bilinear ramp fitting method; and (**a5**) residual tidal displacement from the multi-frame mosaic bilinear ramp fitting method.

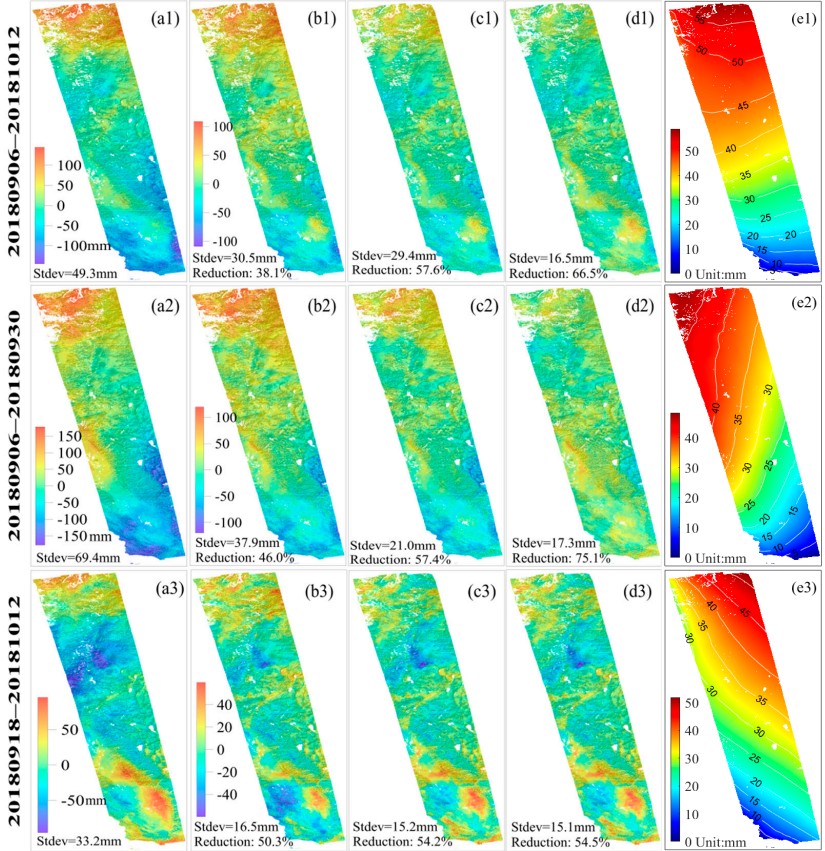

**Figure 8.** (**a1**−**a3**) The long-strip differential interferograms after the correction of (**b1**−**b3**) atmospheric delay error, (**c1**−**c3**) SET and OTL, and (**d1**−**d3**) bilinear ramp elimination. (**e1**−**e3**) The SET and OTL displacements, whose spatial trend variations are similar to the long-strip differential interferograms after the atmospheric delay is corrected.

By analysing the long-wavelength signals of the long-strip differential interferogram in the results in Figure 8, the atmospheric delay is the most vital component in the long-strip differential interferogram, but its magnitude is independent of the spatial scale of the differential interferogram, and the OTL displacement increases with the spatial cover range of the differential interferogram. Therefore, the SET and OTL effects have become the major signals in large-range differential interferograms in coastal areas. The topography-related signal in the long-strip differential interferogram corrected by the atmospheric correction algorithm is weakened, and the StdDev values of the differential interference diagram are reduced by 38.1~50.3% (Figure 8(b1−b3)). The spatial trend signal in the long-strip differential interferogram with the atmospheric delay correction is similar to the tidal displacement of the SET and OTL effects (Figure 8(e1−e3)). Further correction of the SET and OTL displacements further reduces the StdDev value of the differential interferogram by 3.9~19.3% (Figure 8(c1−c3)), and its magnitude depends on the relative spatial variation in the ground tidal displacement. The residual large-scale signal in the long strip differential interferograms is inconsistent with the spatial characteristics of the OTL and approaches the linear plane, which can be eliminated using a bilinear fitting function (Figure 8(d1−d3)). After that, the trend signal in the long-strip differential interferogram is basically eliminated.

The difference between the proposed tidal method and the traditional plane fitting method for OTL displacement correction of the long-strip interferograms is compared. As shown in Figure 9, the pixels of the long-strip differential interferograms were reordered following the tendency of the OTL displacements from small to large, the proposed OTL calculation method can effectively correct the OTL displacement, and the fitting line of the displacement residuals in the interferograms is close to zero and it is basically consistent with that of the PPP displacement residuals, which indicated that the displacement residuals are mainly related to the random noise. Moreover, the original differential interferogram with atmospheric delay error and bilinear ramp correction the OTL corrections in the inland area, but it produces larger errors in the near coastline areas (see Figure 9).

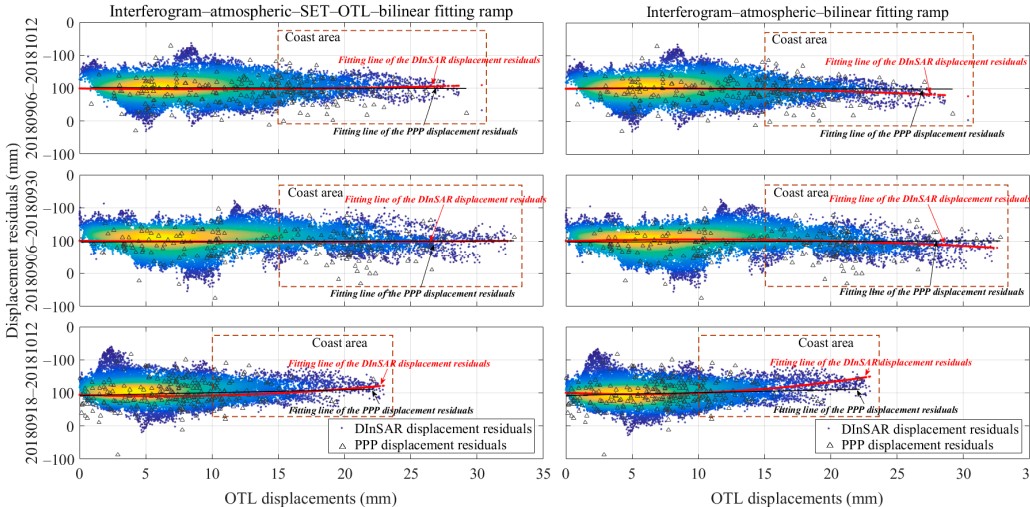

**Figure 9.** The comparison of the displacement residuals in the long-strip differential interferograms after the OTL and SET displacements corrected using the proposed tidal method (interferograms-atmospheric-SET-OTL-bilinear fitting ramp) and traditional bilinear ramp fitting method (interferograms-atmospheric-bilinear fitting ramp).

To further compare and analyse the OTL correction difference between the proposed method and the FES2014b+osu.usawest model, the coastal area A with a larger OTL displacement difference in Figure 4 was selected to validate the improvement of the proposed tidal method. As the results are shown in Figure 10, the absolute value of the fitting line slope of the proposed OTL calculation method is smaller than that of the tidal model in

the interferograms 20180906–20181012 and 20180918–20181012, so the tendency of the OTL displacement can be effectively eliminated that suggests the OTL correction improvement of the proposed tidal method in complex coastline areas.

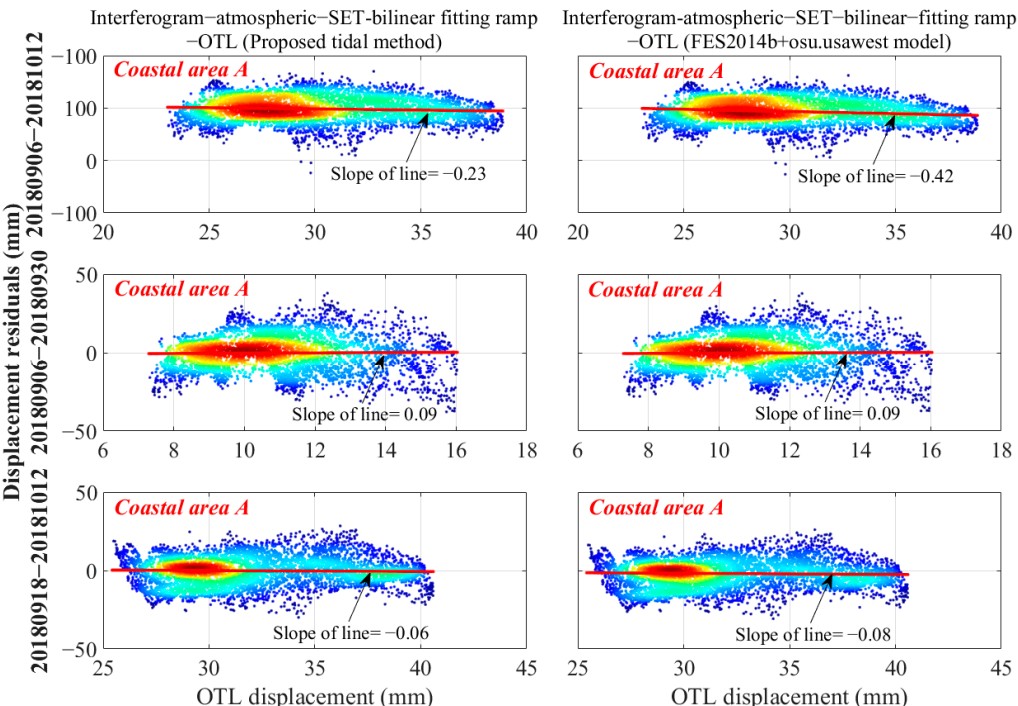

**Figure 10.** The comparison of the displacement residuals in a complex coastline area of the long-strip differential interferograms after the tidal displacements was corrected using the proposed tidal method and FES2014b+osu.usawest model.

## 4. Discussion

We evaluated the ground tidal displacement in the long-strip differential interferogram based on the tidal estimation of the GPS continuous reference station network and the tidal model. For inland areas more than 200 km away from the coastline of the west coast of the U.S., the error produced by bilinear ramp fitting is much less than in coastal areas, which has limited influence on the studies of InSAR ground deformation monitoring; For most coastal areas, there is little difference between the traditional ocean tide model and PPP tidal displacements, and previous studies have proved the OTL displacements in the InSAR measurements can be corrected based on a global ocean tide model [12,13]. However, Figure 4 and Abbaszadeh et al. both indicated that in some localized coastal areas, such as Area A, the maximum StdDev differences between the tidal constituents' displacement time series of recent global ocean tide models are 1~2 mm [22], which may be introduced tidal displacement residuals into the OTL correction of the InSAR ground deformation measurements. To solve this problem, this paper has shown that the PPP tidal displacements can further improve the spatial accuracy of OTL displacements in Area A, and the residual tidal displacement of a few millimetres can be eliminated with the comparison of the commonly used FES2014b+osu.usawest model as shown in Figure 10.

For the long-strip DInSAR measurements, the magnitude and nonlinear variation of tidal displacement are related to the size of the interferogram, shape of the coastline, satellite flight direction, incident angle and imaging time. The OTL effect on the differential interferogram with wider coverage and complex coastline shape was analysed above (see Figures 8–10). In addition to improving the spatial accuracy, the advantages of the method in this paper also can quickly predict the tidal displacement at any time and in any direction according to the system parameters of SAR platforms. If there are dense GPS continuous reference stations in the coastline area that has a larger inter-model discrepancy, such as

coastal areas of Western Europe and East Asia, the spatial accuracy of OTL corrections may be improved by integrating the tidal constituent of the GPS network and tidal models, which will be validated in future studies.

## 5. Conclusions

In this study, we presented a tidal displacement calculation method for long-strip differential interferograms, which integrates tide displacement of the GPS network composed of 1038 continuous sites, the FES2014+osu.usawest model and the SET model in the 2010 IERS Convention. Based on the long-strip differential interferogram of Sentinel-1 SLC images on the west coast of the U.S., the experimental results show that (1) according to the spatial variations of the phasor, the tidal constituent parameters estimated from the PPP coordinate time series of the GPS reference station network can effectively validate the spatial accuracy of the OTL displacement along the LOS direction in the inland area and improve the OTL estimations in the complex coastline area by 10.4%. (2) The SET and OTL effects become the main spatial large-scale signals in the long-strip differential interferograms, the superposition displacement of SET and OTL effects can reach 77.5 mm, the tidal displacement residuals generated by bilinear ramp fitting can reach 20.3 mm, and that generated by the splice-frame bilinear ramp fitting method can reach 7.2 mm with displacement shift where the boundary is mosaicked. (3) The proposed tidal displacement method can effectively eliminate the tendency of tidal displacement in complex coastline areas.

**Author Contributions:** Conceptualization, W.P. and Q.W.; methodology, W.P.; software, W.P.; validation, W.P., Q.W., Y.C., X.X. and W.H.; formal analysis, W.P.; investigation, W.P.; resources, W.P.; data curation, W.P. and Y.C.; writing—original draft preparation, W.P.; writing—review and editing, Q.W., Y.C., X.X. and W.H.; visualization, W.P., X.X. and W.H.; supervision, Q.W.; project administration, Q.W.; funding acquisition, Q.W. All authors have read and agreed to the published version of the manuscript.

**Funding:** This research was funded by Hunan Key Laboratory of remote sensing of ecological environment in Dongting Lake Area (No. 2021-010) of China, Scientific research projects funded by the Department of education of Hunan Province of China (No. 21A0006), Natural Science Foundation of Hunan Province, China (No. 2022JJ40472), and Open Fund of Hunan International Scientific and Technological Innovation Cooperation Base of Advanced Construction and Maintenance Technology of Highway (Changsha University of Science & Technology) (No. kfj210802).

**Data Availability Statement:** The GPS data were provided by Scripps Orbit and Permanent Array Center (SOPAC) at http://sopac-csrc.ucsd.edu (accessed on 15 October 2020). The Sentinel-1 dataset was provided by European Space Agency (ESA) at https://scihub.copernicus.eu (accessed on 6 November 2020). The FES2014 model was available online https://www.aviso.altimetry.fr/en/data/products/auxiliary-products/global-tide-fes.html (accessed on 6 February 2020).

**Acknowledgments:** The NLOADF program was used to produce the OTL correction, and the solid earth tide program was provided by Dehant. The atmospheric delay maps were provided by the ICAMS advanced atmospheric correction method based on the ECMWF ERA-5 global atmospheric model.

**Conflicts of Interest:** The authors declare no conflict of interest.

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
