# Peer review of "Evaluation of Tidal Effect in Long-Strip DInSAR Measurements Based on GPS Network and Tidal Models"

_remotesensing, doi:10.3390/rs14122954_

Round 1

Reviewer 1 Report

Authors have presented a manuscript to evaluate the tidal effect in radar data based on GPS points and tidal model. I found the paper very interesting and with the introduction and methods to be very well explained. However, within materials and methods section I am missing a sub-section about the radar data (in this case Sentinel-1 SLC) and a more precise InSAR processing rather than the one is presented in the results section (it shouldn't be here). Also, the authors should discuss their findings with current literature and I haven't found any discussion in the manuscript.
I would recommend authors to address these concerns before considering the manuscript for publication. 

Reviewer 2 Report

Thank you for submitting your manuscript, and I believe your research topic is important for SAR interferometry for large scale. It is important to understanding and correcting tidal displacement in the differential interferometry, and this manuscript would be helpful and useful for the precise displacement measurement in the future. 

My major concern in your manuscript is that your evaluation is mostly conducted based on your/traditional models such as standard deviation and residuals, and I am not certain whether your estimation of the tidal displacement and the proposed method to correct SAR displacements could provide the reasonable results with the actual physical phenomenon. I believe that it is necessary to compare your result with some actual values from the observed measurements. One suggestion is that your have GPS data, so you can calculate LOS-diplacements that should corresponds to the Sentinel-1 diplacements. Also you can estimate the displacements of SET and OTL at each GPS point. Then, you may select some GPS points and compare the corrected LOS displacements of Sentinel-1 and GPS.

Minor comments:
Line 53: 
Any reference or background information of "FES2004 model"?

Line 73:
It is difficult to understand the variables (M2, N2, O1, Q1) of the model in the introduction. Can you replace them with general description?

Line 78:
Is "three-dimensions" a unique feature for tidal displacement? There is no explanation about the dimensions of the tidal displacement before this sentence. If "three-dimensions" is a highlight of your work, please explain and provide the background information.

Line 170:
The legend title is "StdDev/mm" in Figure1. If this is not unit, please avoid using "/", which is usually meaning division. Also same as other figures.

Line 213-217:
I am not clear why the accuracy of "FES2014b+osu.usawest" model is improved according to the values of the difference and standard deviation in theses sentences. Could you explain the reason in details?

Line 324:
It is difficult to understand these figures without explaining what vertical axis indicates. What is the unit?

Other comments:

In conclusion, it is better to explain about the potential of the versatility for your proposed method, i.e)

  • other areas such as Asia and Europe
  • descending images (different observation time)
  • SAR observation mode (strip map, scansar) 
  • sampling interval of SAR data

Reviewer 3 Report

Dear Editor, Dear authors

This research proposed a tidal displacement calculation method for long-strip differential interferograms, considering both the OTL and SET components of tide displacement. The OTL displacement is estimated by the combination of GPS observation and OTL model, and the SET displacement is retrieved by a SET model. Thereafter, this method is applied to the interferograms along the west coast of the U.S.

The topic of this research is very interesting, and the estimation of tidal displacement is important for the long-strip InSAR data processing. I think the main contribution of this research is to improve the accuracy of tidal displacement estimation. However, the performance validation of this manuscript is not solid enough. 

For example, The StdDev maps in Figure 3 can only show the difference between the two results, rather than the improvement of the proposed method as stated in Line 213-215, since we do not know which result is more accurate without prior information. I suggest to use part of the GPS data for the tidal displacement estimation, and use the other part of the GPS data to validate the improvement of the proposed method.

Moreover, Figure 8 does not show the improvement of the proposed method compared with previous tidal displacement estimation method. The author should add a set of subfigures to illustrate the difference between the traditional OTL model and the proposed method. 

Some minor comments are as follows:

Page 2, Line 49-50, “At present” and “currently” are redundant.

Page, Line 73, It is unclear for readers, what do the “M2, N2, O1 and Q1” mean here, before introducing them in the next section.              

Page 4, Line 137 – “deformation displacement”. The meanings of the two words are similar in this article, the authors can delete either one of them.

Page 4, Equation (7&8) - The atmospheric delay error (datmospheric) occurs both in Equation (7) and (8). What is the relationship between the two equations, is dlarge_scale a new component of the deformation(d) in Equation (8)?

Page 4, Line 146 - Normally x and y in Equation (8) denote the rows and columns of SAR image in radar geometry, since some linear ramps are related with the SAR imaging geometry. On the other hand, longitude and latitude data can also be used to estimate the linear ramps, just make sure you do use the longitude and latitude as the input data.

Page 4, Equation (9) – The first three and the last two components in this equation is conflict. The former (a0+a1x+a2y) is used to estimate the linear ramp of the InSAR phase. However, the later (dOTL+dSET) also contains liner ramp components. Thus, it is not appropriate to list the linear ramp, OTL and SET components together in Equation (9). 

Page 4, Equation 10 - The symbols in this equation need to be explained.

Page 5, Line 195-197 - “The integer times of the constituent S2 (12 hours) period coincide with the SAR satellite’s revisit period, with no relative change between multiple imaging times.” The period of constituent S2 is 12 hours, while the SAR revisit period is generally 12 days, why the two periods “coincide”?

Page 6, Figure 2 - Please explain the “PPP tide phasors” and “PPP-model phasors” in this figure. The figure caption is too simple. Please provide all the necessary information to help readers understand what you plot in the figure without the need of referring the article text. This comment hold for all the figures in the manuscript.

Page 7, Line 226 - “The OTL displacement StdDev along the LOS direction (39°) of the ascending (-13°) …” Does “39°” and “-13°” mean heading angle?

Page 8, Figure 6 - What does the abbreviation BFR in the upper right of the figure mean. 

Page 8, Line 258 - Please explain the “splice bilinear ramp fitting” method.

Page 9, Line 273-274. “…and the plane fitting is not”. Does this mean the plane fitting is not implemented in this step?

Page 10, Line 296-297 - “the StdDev values of the differential interference diagram are reduced by 38.1%~50.3%”. Only when assuming the ground displacement during the InSAR observation period is neglectable and the InSAR phase is mainly related to the noise, the reduction of StdDev can indicate the noise mitigation. The author should declare this if the interferograms meet this standard, 

Page 11, Figure 9. The titles of Y-axis are missing.Page 11, What does the color mean? 

Several sentences are not clear and difficult to understood.

Page 4, Line 137-138 - “…and its deformation displacement expression of the pixel (r, z) in the LOS direction of the differential interferogram can be expressed as…”.

Page 9 Line 284-288 - “Using the atmospheric delay, SET and OTL correction models to correct the atmospheric delay error (especially for long-wavelength and topography-related atmospheric delay errors) and tidal displacements in the long-strip differential interferograms gradually and then bilinear ramp fitting was performed to eliminate the large-scale error residuals”

Page 10, Line 312-315 - “Compared with traditional plane fitting methods, the differential interferograms with the atmospheric delay correction are fitted by the bilinear ramp fitting method, after which the pixel displacement in the long-strip differential interferogram after correcting the tidal displacement by the two methods is analyzed.”

Round 2

Reviewer 1 Report

Thank you very much for addressing the comments.

Although there isn't still discussion between your findings with current literature, so please try to discuss the results with current literature. 

I haven't seen the section 2.4 tidal data analysis and processing in the reviewed version of the manuscript. Please, don't forget to add it.

Reviewer 2 Report

Thank you for your dedication for revising the manuscript. The explanations to my questions are clear enough, and now the revised manuscript is ready for publication.

I look forward to the next phase of the contents in this paper.

Author Response

Thank you for your support and approval, we will do further study on these contents in our paper and extend the proposed tidal calculation method in other localized coastal areas with a larger inter-model discrepancy.

Reviewer 3 Report

Authors made necessary changes for the manuscript, and response my concerns accordingly. Only a minor thing:

Point 3: Page, Line 73, It is unclear for readers, what do the M2, N2, O1 and Q1 mean here, before introducing them in the next section. 

Here, I mean, please explain what M2, N2, O1 and Q1 represent for, i.e., why we called M2, N2, O1 and Q1 in the harmonic analysis method?

Author Response

Point 3: Page, Line 73, It is unclear for readers, what do the “M2, N2, O1 and Q1” mean here, before introducing them in the next section. 

Here, I mean, please explain what M2, N2, O1 and Q1 represent for, i.e., why we called M2, N2, O1 and Q1 in the harmonic analysis method?

Response 3: We are extremely grateful to the reviewer for pointing out this problem. The OTL displacement can generally be decomposed into semi-diurnal constituents (M2, N2, S2, K2) and diurnal constituents (Q1, O1, P1, K1). 

M2 represents the semi-diurnal principal lunar constituent, N2 is the semi-diurnal elliptical lunar constituent, O1 is the diurnal principal lunar constituent and Q1 is the diurnal elliptical lunar constituent.

The symbols for the main constituents were designed by George Darwin in the 1890s to distinguish the tide constituents of different frequencies.

The sentence in the original manuscript “The OTL displacement can be decomposed into several constituents according to the harmonic analysis method, and the GPS precise point positioning (PPP) technique has been shown to accurately determine the displacement of M2, N2, O1 and Q1 constituents in north (N), east (E) and up (U) directions [21-23]”

have been corrected as:

“The OTL displacement can generally be decomposed into eight main tide constituents (semi-diurnal constituents M2, N2, S2, K2 and diurnal constituents Q1, O1, P1, K1) according to the harmonic analysis method, and the GPS precise point positioning (PPP) technique has been shown to accurately determine the tide constituents displacement of semi-diurnal principal lunar M2, semi-diurnal elliptical lunar N2, diurnal principal lunar O1 and diurnal elliptical lunar Q1 in the north (N), east (E) and up (U) directions [21-23]”.